# Global, Regional, and National Burdens with Temporal Trends of Early-, Intermediate-, and Later-Onset Gastric Cancer from 1990 to 2019 and Predictions up to 2035

**DOI:** 10.3390/cancers14215417

**Published:** 2022-11-03

**Authors:** Fei-Long Ning, Nan-Nan Zhang, Zhe-Ming Zhao, Wan-Ying Du, Yong-Ji Zeng, Masanobu Abe, Jun-Peng Pei, Chun-Dong Zhang

**Affiliations:** 1Department of Gastrointestinal Surgery, The Fourth Affiliated Hospital, China Medical University, Shenyang 110032, China; 2Department of Gastrointestinal Surgery, Graduate School of Medicine, The University of Tokyo, Tokyo 113-8655, Japan; 3National Clinical Research Center and State key Laboratory of Cancer Biology for Digestive Diseases, Xijing Hospital of Digestive Diseases, Fourth Military Medical University, Xi’an 710000, China; 4Section of Gastroenterology, Department of Medicine, Baylor College of Medicine, Houston, TX 77030, USA; 5Division for Health Service Promotion, The University of Tokyo, Tokyo 113-8655, Japan; 6Key Laboratory of Carcinogenesis and Translational Research (Ministry of Education/Beijing), Peking University Cancer Hospital & Institute, Beijing 100142, China

**Keywords:** epidemiology, gastric cancer, temporal trend, prediction

## Abstract

**Simple Summary:**

Early, intermediate, and late-onset gastric cancer (EOGC, IOGC, LOGC) incidence and death rates differed globally, regionally, and nationally in 2019. From 1990 to 2019, EOGC showed a slower decrease in incidence rate worldwide than IOGC and LOGC, whereas EOGC and LOGC showed slower decreases in mortality than IOGC. The worldwide incidence rate of EOGC was predicted to increase substantially from 2020 to 2035, while that for LOGC was predicted to increase slightly and that for IOGC was predicted to remain stable over the same period. This study revealed significant differences in the burdens and temporal trends of EOGC, IOGC, and LOGC, and highlighted the importance of tailored cancer-control measures in neglected subpopulations, especially in patients with EOGC.

**Abstract:**

Background: Evidence for estimating and predicting the temporal trends of gastric cancer in different age groups is lacking. Methods: Data of early-, intermediate-, and later-onset gastric cancer (EOGC, IOGC, LOGC) was from the Global Burden of Diseases Study 2019. The incidences and deaths due to EOGC, IOGC, and LOGC were analyzed by period, sex, geographic location, and sociodemographic incidence. Temporal trends were evaluated by estimated annual percentage changes (EAPCs). The incidences and temporal trends were predicted until 2035. Results: There were substantial differences in the incidence and death rates of the three populations at global, regional and national levels in 2019. From 1990 to 2019, EOGC (EAPC, −0.84) showed a slower decrease in incidence rate worldwide than IOGC (EAPC, −1.77) and LOGC (EAPC, −1.10), whereas EOGC and LOGC showed slower decreases in mortality than IOGC. The worldwide incidence rate of EOGC (EAPC, 1.44) was predicted to increase substantially from 2020 to 2035, while that for LOGC (EAPC, 0.43) was predicted to increase slightly and that for IOGC (EAPC, −0.01) was predicted to remain stable over the same period. Conclusions: This study revealed differences in the burdens and temporal trends of EOGC, IOGC, and LOGC, and highlighted the importance of tailored cancer-control measures in neglected subpopulations, especially in patients with EOGC.

## 1. Introduction

Gastric cancer remains an important cancer, with the fifth-highest incidence and fourth-highest mortality globally [1]. Despite steady declines in the incidence rate of gastric cancer over the last half century in many parts of the world, the incidence of early-onset gastric cancer (EOGC), diagnosed at age < 50 years, has been predicted to increase in both low- and high-risk countries in the next decade [2,3]. The reasons for the rise among young adults remain unknown but may be related to the rising prevalence of autoimmune gastritis, dysbiosis of the gastric microbiome, and the increased use of antibiotics among younger generations [4,5]. This rise in EOGC, together with various early-onset cancers in other organs, has attracted growing global concern [6].

Whether gastric cancer among younger adults represents a distinct disease from that in elderly adults is an important clinical issue. Gastric cancer among younger adults appears to be genetically and clinically distinct from traditional gastric cancer, and is associated with a more-diffuse histology, poor differentiation, and more peritoneal metastasis. Differences in the incidence and death rates among gastric cancers in different age groups are also unclear; for example, EOGC diagnosed at age <50 years, intermediate-onset gastric cancer (IOGC) diagnosed at age 50–54 years, and later-onset gastric cancer (LOGC) at age ≥55 years. Furthermore, the incidence and death rates of EOGC, IOGC, and LOGC have not been systematically estimated and predicted, especially at regional and national levels [7,8].

Based on the Global Burden of Disease (GBD) Study 2019 [9,10], we analyzed the incidences, deaths, and temporal trends of EOGC, IOGC, and LOGC from 1990 to 2019 at global, regional, and national levels in relation to period, sex, geographic location, and sociodemographic index (SDI). We also predicted the future incidences, deaths, and temporal trends at global and national levels from 2020 to 2035.

## 2. Materials and Methods

### 2.1. Definitions of EOGC, IOGC, and LOGC

All cases coded as C16–C16.9, Z12.0, and Z85.02–Z85.028 in the International Classification of Diseases 10th Revision (ICD–10) were considered to be gastric cancer [10,11]. In this cross-sectional study, EOGC was defined as gastric cancer diagnosed at age 15–49 years, IOGC as gastric cancer diagnosed at age 50–54 years, and LOGC as gastric cancer diagnosed at age ≥ 55 years.

### 2.2. Data Sources

Based on geographical locations, the GBD 2019 was grouped into 21 regions or 203 countries. All countries were further grouped into low-, low-middle-, middle-, high-middle-, and high-SDI regions [12]. The burdens and temporal trends in EOGC, IOGC, and LOGC were analyzed in relation to the 21 GBD regions, five SDI regions, and 203 countries, respectively. We extracted data from the GBD 2019 Data Resources (https://ghdx.healthdata.org/gbd-2019) (accessed on 18 January 2022) between 1990 and 2019 using the Global Health Data Exchange tool (GHDx).

This study was approved by the Institutional Ethics Committees of The Fourth Affiliated Hospital of China Medical University (EC–2021–KS–068) and was performed according to the guidelines of the Declaration of Helsinki. This study was reported according to the Strengthening the Reporting of Observational Studies in Epidemiology (STROBE) guidelines [13].

### 2.3. Statistical Analysis

The incidences, deaths, and corresponding rates of EOGC, IOGC, and LOGC between 2019 and 1990 were estimated at global, regional, and national levels. The estimated annual percentage change (EAPC) was calculated to quantify time trends in incidence and death rates using Joinpoint software (version 4.7.0.0; National Cancer Institute, Rockville, MD, USA) and R version 4.1.2 (R Foundation for Statistical Computing, Vienna, Austria).

The expected relationships between SDI values and incidence and death rates were determined by fitting a Gaussian process regression to estimates at global, regional, and national levels from 1990 to 2019. The correlation between the EAPCs for incidence and death rates and SDI values in 2019 were further evaluated by Pearson’s correlation analyses.

Based on the GBD data from 2004 to 2019, we applied a Bayesian age–period–cohort model to predict the incidence rates from 2020 to 2035 at worldwide and national levels. The quality of data for each country was classified from 0–5 stars, as described previously [14,15]. We selected 12 study countries with high-quality data (3–5 stars).

Further details are provided in the Appendix A.

## 3. Results

### 3.1. Burdens of EOGC, IOGC, and LOGC in 2019 vs. 1990

#### 3.1.1. Global Burdens

Globally, there were 144,321 incident cases of EOGC in 2019 compared with 125,974 cases in 1990, with incidence rates of 3.7 and 4.6 per 100,000, respectively (Table 1; Figure 1A). For IOGC, there were 92,619 incident cases in 2019 and 74,469 cases in 1990, with respective incidence rates of 21.2 and 35.0 per 100,000. There were 1,032,865 incident cases of LOGC in 2019 and 682,952 cases in 1990, and the incidence rates were 101.8 and 73.5 per 100,000, respectively.

The global numbers of deaths due to EOGC were 87,333 in 2019 and 97,383 in 1990, corresponding to death rates of 2.2 and 3.6 per 100,000, respectively (Appendix A; Figure 1B). The corresponding figures for IOGC were 60,169 deaths in 2019 and 60,688 in 1990, with respective death rates of 13.8 and 28.5 per 100,000, and the numbers for LOGC were 809,683 deaths in 2019 compared with 630,247 in 1990, with death rates of 57.6 and 93.9 per 100,000, respectively. The incidence and death rates in relation to sex are presented in Appendix A.

#### 3.1.2. Regional Burdens

In 2019, the highest incidence rate of EOGC was in East Asia (9.2 per 100,000), followed by high-income Asia Pacific, and the highest death rate was also found in East Asia (4.3 per 100,000), followed by Eastern Europe (Table 1; Figure 1). The highest incidence rate of IOGC was also in East Asia (39.5 per 100,000), followed by high-income Asia Pacific, while the highest death rate was in East Asia (22.1 per 100,000), followed by Central Asia. The highest incidence rate of LOGC was in high-income Asia Pacific (169.8 per 100,000), followed by East Asia, whereas the highest death rate was in Andean Latin America (106.4 per 100,000), followed by East Asia. The highest incidence and death rates of EOGC, IOGC, and LOGC in 2019 were all in high-middle SDI regions in 2019. The incidence and death rates in the 21 GBD regions in relation to sex are presented in Appendix A.

#### 3.1.3. National Burdens

The burdens of EOGC, IOGC, and LOGC varied widely among countries and between sexes (Figure 2 and Appendix A). In 2019, the highest incidence rate of EOGC was in the Solomon Islands (10.7 per 100,000), followed by Mongolia, and the highest death rate was in Mongolia (9.3 per 100,000), followed by the Solomon Islands. For IOGC, the highest incidence rate (64.1 per 100,000) and death rate (58.0 per 100,000) were in Mongolia, followed by Afghanistan, while the highest incidence rate of LOGC was in Japan (185.7 per 100,000), followed by Mongolia, and the highest death rate was in Mongolia (181.1 per 100,000), followed by Bolivia.

### 3.2. Temporal Trends of EOGC, IOGC, and LOGC, 1990–2019

#### 3.2.1. Global Trends

From 1990 to 2019, the incidence rates of EOGC (EAPC, −0.84; 95% CI, −0.99 to −0.69), IOGC (EAPC, −1.77; 95% CI, −2.03 to −1.50), and LOGC (EAPC, −1.10; 95% CI, −1.25 to −0.95) showed overall decreasing trends worldwide, and the death rates of EOGC (EAPC, −1.78; 95% CI, −2.00 to −1.57), IOGC (EAPC, −2.55; 95% CI, −2.85 to −2.26) and LOGC (EAPC, −1.69; 95% CI, −1.88 to −1.51) also presented overall decreasing trends in the same period (Table 1 and Appendix A).

#### 3.2.2. Regional Trends

The incidence and death rates of EOGC, IOGC, and LOGC decreased in all five SDI regions from 1990 to 2019 (Table 1 and Appendix A). The greatest decreases in incidence rates of EOGC (EAPC, −2.41; 95% CI, −2.49 to −2.32), IOGC (EAPC, −3.24; 95% CI, −3.33 to −3.15), and LOGC (EAPC, −1.73; 95% CI, −1.80 to −1.65) occurred in high-SDI regions, while the death rates (EOGC: EAPC, −3.31; 95% CI, −3.40 to −3.22; IOGC: EAPC, −3.81; 95% CI, −3.94 to −3.68; LOGC: EAPC, −2.16; 95% CI, −2.23 to −2.10) also decreased most in high-SDI regions. The incidence and death rates in SDI regions during specific periods are presented in Appendix A.

From 1990 to 2019, the incidence and death rates of EOGC showed overall downward trends in most GBD regions (Table 1 and Appendix A). The incidence rate of EOGC decreased most from 1990 to 2019 in high-income Asia Pacific (EAPC, −3.08; 95% CI, −3.23 to −2.92), followed by Eastern Europe, and increased most in East Asia (EAPC, 0.98; 95% CI, 0.69−1.27). The death rate of EOGC also decreased most in high-income Asia Pacific (EAPC, −4.45; 95% CI, −4.62 to −4.28), followed by Eastern Europe, with only Oceania (EAPC, 0.41; 95% CI, 0.30–0.51) showing an upward trend from 1990 to 2019. The incidence and death rates of IOGC and LOGC showed overall declining trends from 1990 to 2019 in all regions. Both the incidence rate (EAPC, −3.80; 95% CI, −3.97 to −3.64) and death rate (EAPC, −4.75; 95% CI, −4.88 to −4.62) of IOGC decreased most in high-income Asia Pacific, followed by Eastern Europe, while the greatest decreasing trend in the incidence rate of LOGC was in Eastern Europe (EAPC, −2.62; 95% CI, −2.81 to −2.42), followed by tropical Latin America, and the largest decreasing trend in death rate was in high-income North America (EAPC, −2.37; 95% CI, −2.51 to −2.22), followed by Western Europe. The temporal trends in incidence and death rates of EOGC, IOGC, and LOGC in relation to SDI and GBD regions during specific periods are shown in Appendix A.

#### 3.2.3. National Trends

We further analyzed the temporal trends of incidence and death rates of EOGC, IOGC, and LOGC in 203 countries from 1990 to 2019, and substantial differences were also found at the national level (Appendix A).

### 3.3. Impact of SDI on Incidence and Death Rates, 1990–2019

The relationship between the global- and regional-level incidence and death rates and SDI values are shown in Appendix A. At the global level, the incidence and death rates of EOGC, IOGC, and LOGC all showed decreasing trends with increasing SDI values. The incidence and death rates of EOGC showed decreasing trends with increasing SDI values in most regions; however, there were variations, with Western sub-Saharan Africa and Eastern Europe showing the largest decreases in incidence rates with increasing SDI values, and East Asia and Oceania showing increasing trends in incidence rates. Similarly, Western sub-Saharan Africa and Eastern Europe showed the largest decreases in death rates with increasing SDI values, and only Oceania showed an increasing trend. The incidence and death rates of IOGC and LOGC in all regions showed decreasing trends with increasing SDI values. The relationships between SDI values and incidence and death rates in relation to sex are shown in Appendix A.

The relationships between incidence and death rates and SDI values at the national level are shown in Appendix A. There were large differences; for example, China and the Republic of Korea had much higher incidence rates than expected for EOGC, IOGC, and LOGC, whereas South Africa and Nigeria had lower values than expected based on SDI values. Similarly, the death rates due to EOGC, IOGC, and LOGC in China and the Republic of Korea were much higher than expected, while South Africa and Nigeria had lower values than expected based on SDI.

We further analyzed the association between EAPC and SDI values (Figure 3). There was no significant correlation between the EAPC of incidence rate for EOGC and SDI values (ρ = −0.1, *p* = 0.15). However, the EAPCs of incidence rates for IOGC and LOGC were negatively associated with SDI values (ρ = −0.42, ρ = −0.42, respectively; both *p* < 0.001) (Figure 3A), while the EAPCs of deaths rates for EOGC, IOGC, and LOGC were also negatively correlated with SDI values (ρ = −0.35, ρ = −0.45, ρ = −0.54, respectively; all *p* < 0.001) (Figure 3B).

### 3.4. Predicted Incidences from 2020 to 2035

The incidences and incidence rates of EOGC, IOGC, and LOGC were predicted worldwide and in the 12 studied countries from 2020 to 2035 (Figure 4; Appendix A). The incidence of EOGC was predicted to almost double in 2035 compared with 2019 in China and Korea, while a slight decrease was predicted in Japan. The global incidence rate of EOGC was expected to increase from 3.7 per 100,000 in 2021 to 4.6 per 100,000 in 2035 (Figure 4A). Notably, the incidence rates in China and Korea were predicted to increase substantially, from 9.2 and 10.9 per 100,000 in 2019 to 17.6 and 21.4 per 100,000 in 2035, respectively, while the incidence rate in Japan was expected to decline to 6.6 per 100,000 in 2028 and then rise again to 7.1 per 100,000 in 2035. The incidence rates in Australia and the United States of America were predicted to change only slightly or remain stable.

The incidence of IOGC was predicted to increase in Korea and the USA, and to decrease slightly or remain stable in Japan and France (Appendix A), while the global incidence rate was predicted to decrease from 21.1 per 100,000 in 2019 to 18.8 per 100,000 in 2027, and then increase again to 21.1 per 100,000 in 2035 (Figure 4B). Similarly, the incidence rate in China was expected to decrease from 41.2 per 100,000 in 2019 to 36.8 per 100,000 in 2025, and then increase significantly to >59 per 100,000 in 2035, while the incidence rate in Korea was predicted to increase substantially from 9.2 per 100,000 in 2019 to 21.4 per 100,000 in 2035. The incidence rates in Japan and South Africa were predicted to decrease rapidly from 29.9 and 7.7 per 100,000 in 2019 to 24.5 and 2.3 per 100,000 in 2035, respectively.

The number of new LOGC cases was expected to increase in most of the studied countries (Appendix A). Globally, the incidence rate of LOGC was predicted to change slightly, with similar trends in many countries, including Japan, France, and Italy (Figure 4C). However, the incidence rates in China and Korea were predicted to increase sharply from 148.0 and 134.0 per 100,000 in 2019 to 183.9 and 217.6 per 100,000 in 2035, while the incidence rate in Brazil was predicted to decline from 48.1 per 100,000 in 2019 to 41.3 per 100,000 in 2035.

## 4. Discussion

To the best of our knowledge, this study provides the first analyses of the burden, temporal trends, and predicted trends in EOGC, IOGC, and LOGC worldwide, according to GBD regions, SDI regions, and countries, thus providing important epidemiological information. We reported the population-based incidences, deaths, and temporal trends from 1990−2019, and the predictions from 2020−2035 in relation to period, sex, geographic location, and SDI. Furthermore, we analyzed the clinicopathological, somatic, and survival characteristics of patients with EOGC, IOGC, and LOGC.

The estimations from the GBD 2019 were consistent with previous studies. For example, GLOBOCAN estimated 1,033,701 incident cases and 782,685 deaths due to gastric cancer in 185 countries in 2018, while GBD 2019 estimated 1,238,326 cases and 938,560 deaths in 203 countries in 2018 [1]. A total of 390,182 deaths were reported in China in 2018, compared with 412,924 estimated by GBD 2019 [16]. The reported number of gastric cancers in the Republic of Korea in 2018 was 29,279, compared with 25,074 estimated by the GBD 2019 [17]. Overall, the estimations of the GBD 2019 should thus be considered reliable.

Persistent decreasing trends in the incidence and death rates of EOGC, IOGC, and LOGC have been observed globally from 1990 to 2019. These observed declines in the incidence and death from gastric cancer have been considered as “an unplanned triumph” in cancer prevention, given that they occurred in the absence of active primary prevention programs, except for population-based screening in Japan and South Korea [18,19,20]. These decreasing trends can partly be explained by improved food-preservation methods, improved living conditions, economic developments, reduced overcrowding, better hygiene, and advances in *Helicobacter pylori* eradication [21,22,23]. Notably, the decreasing trend in incidence rate during the past three decades was more apparent for IOGC compared with EOGC and LOGC, although the reasons for this difference remain unclear.

Our predictions suggest that the incidence rates of EOGC and LOGC will increase worldwide from 2020 to 2035. However, EOGC is predicted to increase more than LOGC globally, and in both high-risk countries (e.g., China and the Republic of Korea) and low-risk countries (e.g., the USA and United Kingdom). Gastric cancers located in both the cardia and corpus have been increasing, especially in younger generations [4,5] with obesity and gastroesophageal reflux disease considered to be the main risk factors for this increase [2,4,24].

The incidence and death rates of gastric cancer in relation to SDI raise important issues. Notably, consistent with established epidemiological data, reductions in gastric cancer burden occurred mainly in high- and high-middle SDI regions. We accordingly found significant negative correlations between SDI values and temporal trends (EAPCs) in incidence and death rates, with more obvious downward trends in incidence and death rates in countries with higher SDI values. Importantly, low-SDI regions showed similarly low incidence and death rates to high-SDI regions in 2019; however, we cannot exclude the possibility that the apparently low burden in low-SDI regions may be due to underestimations of cancer incidence and deaths in limited-resource settings, due to potential misdiagnosis or missed diagnosis, or underdeveloped cancer register systems [25].

This study had several limitations that should be noted before interpreting the main findings. The accuracy of the results depended on the quality and availability of the data at a given time and in a given country, and the poor availability and quality of data from some low-SDI countries with large rural populations could only be partially overcome by improving the estimation framework. We were also unable to analyze the possible risk factors for EOGC, IOGC, and LOGC due to the unavailability of data from the GBD 2019. In addition, the association between SDI values and the burdens of EOGC, IOGC, and LOGC should be interpreted with caution, given that we did not control for other potential factors due to the unavailability of covariate data. Finally, further studies are warranted to develop a forecast model including possible changes in influencing factors.

## 5. Conclusions

There were substantial differences in the incidence and death rates of EOGC, IOGC, and LOGC at global, regional and national levels in 2019. EOGC showed a slower decrease in incidence rate than IOGC and LOGC globally from 1990 to 2019, whereas EOGC and LOGC showed slower decreases in death rates than IOGC. The incidence rate of EOGC is generally predicted to increase substantially from 2020 to 2035 worldwide, while the rate for LOGC will only increase slightly, and the rate for IOGC is predicted to remain stable over the same period. This study thus revealed variations among gastric cancers in different age groups in relation to period, sex, geographic location, and SDI, and it highlighted the importance of tailored cancer-control measures in neglected subpopulations, especially in patients with EOGC. The results of this study may thus help policymakers to make better public policy decisions.

## Figures and Tables

**Figure 1 cancers-14-05417-f001:**
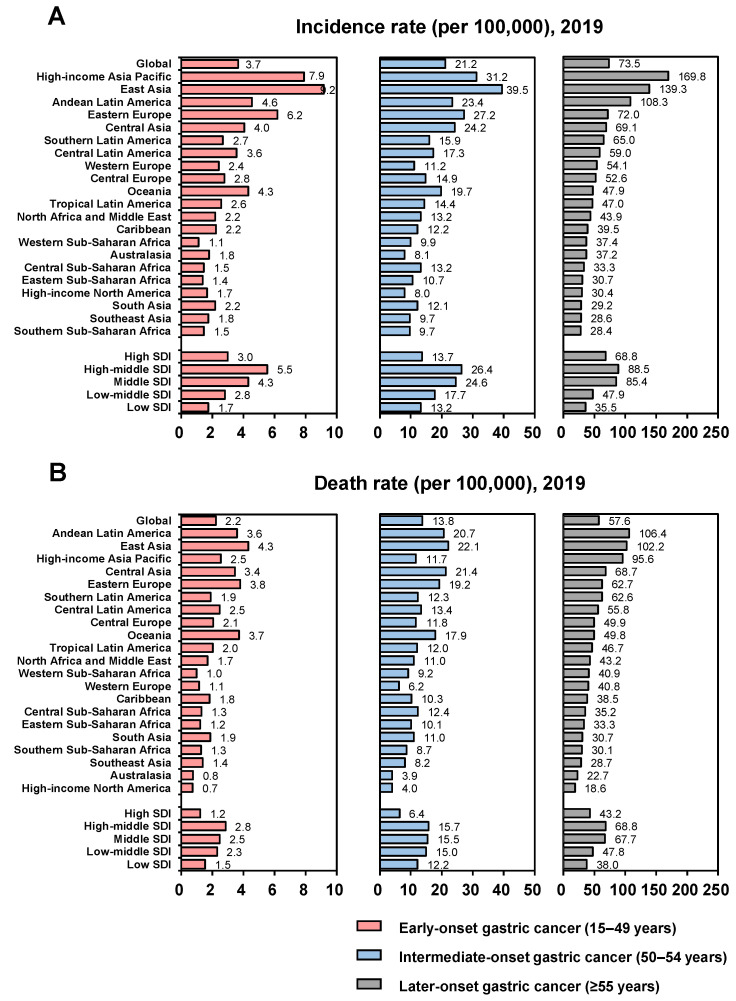
Incidence and death rates globally and by SDI and GBD regions in 2019. (**A**) Incidence rates globally and by SDI and GBD regions in 2019; (**B**) Death rates globally and by SDI and GBD regions in 2019.

**Figure 2 cancers-14-05417-f002:**
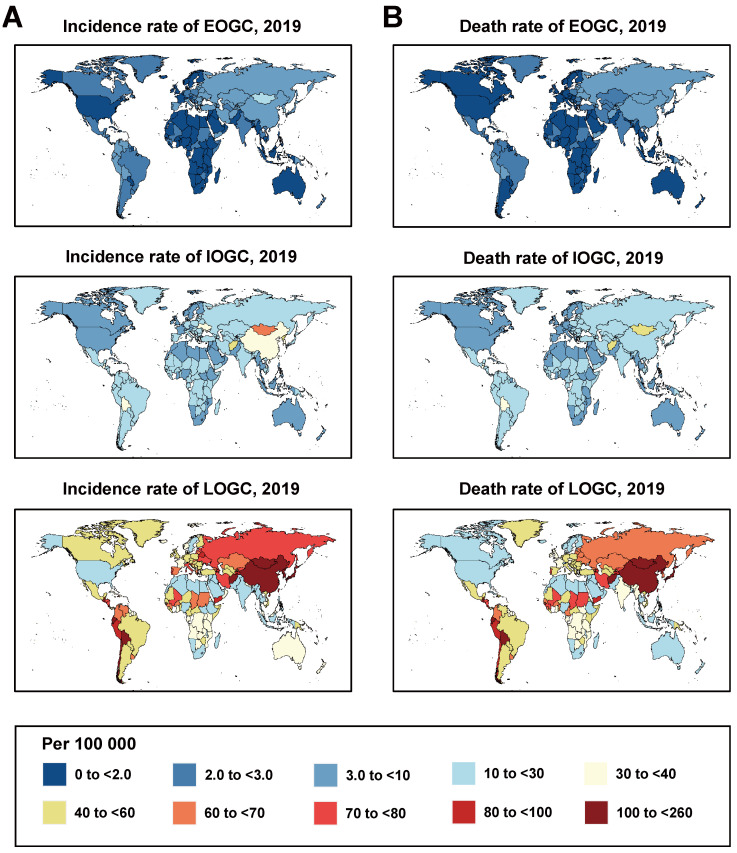
Incidence and death rates of gastric cancer globally in 2019. (**A**) Incidence rates of gastric cancer globally in 2019; (**B**) Death rates of gastric cancer globally in 2019.

**Figure 3 cancers-14-05417-f003:**
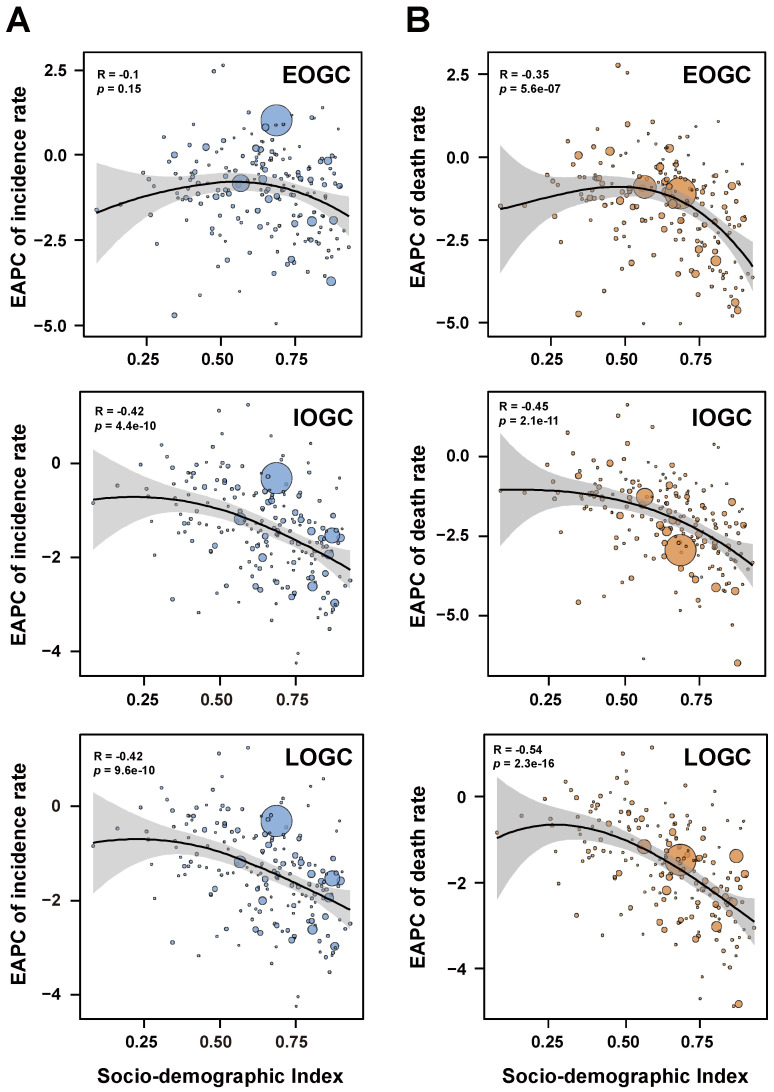
Correlations between estimated annual percentage changes in gastric cancer and SDI values. (**A**) Correlations between estimated annual percentage changes of incidence rates in gastric cancer and SDI values; (**B**) Correlations between estimated annual percentage changes of death rates in gastric cancer and SDI values.

**Figure 4 cancers-14-05417-f004:**
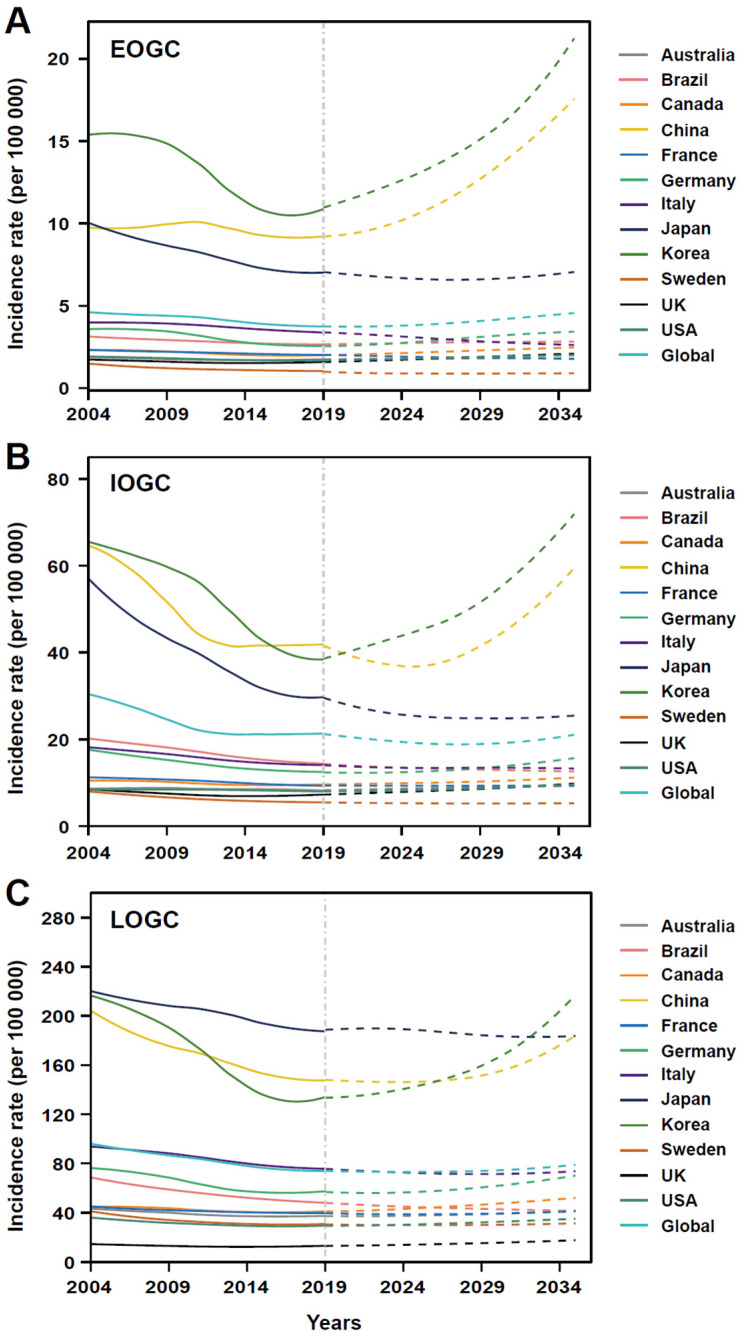
Predictions for gastric cancer from 2020 to 2035 globally and in 12 countries. (**A**) Predictions of incidence rate for early-onset gastric cancer from 2020 to 2035 globally and in 12 countries; (**B**) Predictions of incidence rate for intermediate-onset gastric cancer from 2020 to 2035 globally and in 12 countries; (**C**) Predictions of incidence rate for later-onset gastric cancer from 2020 to 2035 globally and in 12 countries.

**Table 1 cancers-14-05417-t001:** Incidences and temporal trends in gastric cancers globally and by sex, and SDI and GBD regions, 1990–2019.

Population	1990		2019		1990–2019
	Incidence Cases	IR (per 100,000)	Incidence Cases	IR (per 100,000)	EAPC
	No. (95% UI)	No. (95% UI)	No. (95% UI)	No. (95% UI)	No. (95% CI)
Global					
EOGC	125,974 (118,140–133,956)	4.6 (4.4–4.9)	144,321 (130,473–158,789)	3.7 (3.3–4.0)	−0.84 * (−0.99 to −0.69)
IOGC	74,469 (69,424–79,643)	35.0 (32.7–37.5)	92,619 (81,908–104,377)	21.2 (18.8–23.9)	−1.77 * (−2.03 to −1.50)
LOGC	682,952 (645,714–718,165)	101.8 (96.2–107.0)	1,032,865 (934,144–1,135,210)	73.5 (66.4–80.8)	−1.10 * (−1.25 to −0.95)
Sex					
Male					
EOGC	73,551 (67,773–79,096)	5.4 (4.9–5.8)	90,805 (79,515–103,618)	4.6 (4.0–5.2)	−0.44 * (−0.65 to −0.24)
IOGC	50,923 (46,691–55,835)	47.3 (43.4–51.9)	64,844 (54,890–76,597)	29.8 (25.2–35.2)	−1.56 * (−1.86 to −1.25)
LOGC	423,665 (395,052–453,227)	136.2 (127.0–145.7)	691,222 (610,582–782,032)	104.4 (92.2–118.1)	−0.85 * (−1.00 to −0.69)
Female					
EOGC	52,423 (47,705–57,109)	3.9 (3.6–4.3)	53,516 (47,850–59,347)	2.8 (2.5–3.1)	−1.45 * (−1.61 to −1.29)
IOGC	23,545 (21,572–25,377)	22.4 (20.6–24.2)	27,775 (24,563–31,158)	12.7 (11.2–14.2)	−2.15 * (−2.36 to −1.93)
LOGC	259,287 (241,989–275,659)	72.0 (67.2–76.6)	341,643 (301,534–380,296)	45.9 (40.5–51.1)	−1.62 * (−1.77 to −1.47)
Sociodemographic index (SDI)					
High SDI					
EOGC	24,633 (24,089–25,167)	5.7 (5.6–5.9)	14,042 (12,929–15,275)	3.0 (2.8–3.3)	−2.41 * (−2.49 to −2.32)
IOGC	14,382 (13,895–14,824)	33.0 (31.8–34.0)	9392 (8471–10,436)	13.7 (12.3–15.2)	−3.24 * (−3.33 to −3.15)
LOGC	188,546 (180,270–192,729)	107.4 (102.6–109.7)	214,329 (187,179–235,870)	68.8 (60.1–75.7)	−1.73 * (−1.80 to −1.65)
High-middle SDI					
EOGC	35,030 (32,515–37,646)	5.8(5.4–6.2)	40,349 (35,304–45,650)	5.5 (4.8–6.3)	−0.34 * (−0.51 to −0.16)
IOGC	24,921 (23,304–26,742)	44.1 (41.3–47.4)	26,927 (22,777–31,567)	26.4 (22.3–31.0)	−1.75 * (−2.08 to −1.41)
LOGC	231,643 (218,770–244,500)	125.1 (118.2–132.0)	313,798 (278,349–351,211)	88.5 (78.5–99.0)	−1.17 * (−1.37 to −0.97)
Middle SDI					
EOGC	41,707 (37,520–46,082)	4.6 (4.2–5.1)	54,282 (47,259–61,912)	4.3 (3.7–4.9)	−0.18 (−0.42 to 0.06)
IOGC	23,115 (20,358–26,278)	38.0 (33.5–43.2)	37,838 (31,791–44,635)	24.6 (20.7–29.1)	−1.53 * (−1.93 to −1.13)
LOGC	186,596 (168,262–207,174)	108.4 (97.7–120.3)	366,101 (319,384–48,858)	85.4 (74.5–97.7)	−0.64 * (−0.89 to −0.39)
Low-middle SDI					
EOGC	18,731 (16,680–20,404)	3.5 (3.1–3.8)	26,153 (23,536–28,918)	2.8 (2.5–3.1)	−0.70 * (−0.80 to −0.59)
IOGC	9267 (8327–10,233)	25.2 (22.6–27.8)	14,138 (12,502–15,857)	17.7 (15.7–19.9)	−1.29 * (−1.41 to −1.17)
LOGC	57,799 (53,087–62,058)	58.4 (53.6–62.7)	108,896 (99,437–118,824)	47.9 (43.7–52.3)	−0.66 * (−0.76 to −0.57)
Low SDI					
EOGC	5836 (4921–6622)	2.5 (2.1–2.8)	9442 (8062–10,865)	1.7 (1.5–2.0)	−1.28 * (−1.30 to −1.25)
IOGC	2766 (2379–3127)	18.7 (16.1–21.2)	4296 (3674–4949)	13.2 (11.3–15.2)	−1.35 * (−1.44 to −1.26)
LOGC	18,167 (16,102–20,132)	46.8 (41.5–51.9)	29,451 (26,698–32,482)	35.5 (32.2–39.2)	−0.97 * (−1.02 to −0.92)
GBD regions					
Andean Latin America					
EOGC	985 (889–1096)	5.3 (4.8–5.9)	1511 (1177–1918)	4.6 (3.6–5.8)	−0.62 * (−0.82 to −0.42)
IOGC	429 (373–491)	37.4 (32.5–42.7)	702 (527–922)	23.4 (17.6–30.8)	−1.58 * (−1.71 to −1.44)
LOGC	4561 (4139–4987)	135.7 (123.2–148.4)	10,161 (8347–12,276)	108.3 (89.0–130.9)	−0.77 * (−0.87 to −0.66)
Australasia					
EOGC	211 (198–226)	2.0 (1.8–2.1)	242 (186–310)	1.8 (1.4–2.3)	−0.24 * (−0.35 to −0.13)
IOGC	114 (104–127)	12.0 (10.8–13.3)	147 (106–203)	8.1 (5.8–11.1)	−1.30 * (−1.46 to −1.13)
LOGC	2057 (1953–2147)	52.2 (49.6–54.5)	3060 (2451–3739)	37.2 (29.8–45.4)	−1.47 * (−1.59 to −1.35)
Caribbean					
EOGC	409 (349–453)	2.2 (1.9–2.5)	535 (432–644)	2.2 (1.8–2.7)	0.00 (−0.07 to 0.07)
IOGC	211 (184–242)	16.2 (14.1–18.6)	335 (270–408)	12.2 (9.9–14.9)	−0.69 * (−0.81 to −0.58)
LOGC	2303 (2117–2434)	53.3 (49.0–56.3)	3486 (3023–3982)	39.5 (34.2–45.1)	−0.93 * (−1.04 to −0.82)
Central Asia					
EOGC	2161 (2077–2244)	6.5 (6.2–6.7)	1974 (1765–2230)	4.0 (3.6–4.6)	−2.19 * (−2.40 to −1.98)
IOGC	1618 (1537–1694)	54.4 (51.7–57.0)	1158 (1009–1341)	24.2 (21.1–28.0)	−2.81 * (−2.92 to −2.69)
LOGC	9613 (9274–9907)	120.6 (116.3–124.3)	8994 (8146–9923)	69.1 (62.6–76.2)	−1.82 * (−1.94 to −1.69)
Central Europe					
EOGC	2511 (2452–2570)	4.1 (4.0–4.2)	1460 (1262–1678)	2.8 (2.4–3.2)	−1.85 * (−2.03 to −1.66)
IOGC	1790 (1724–1863)	26.0 (25.0–27.1)	1113 (948–1299)	14.9 (12.7–17.4)	−1.91 * (−2.05 to −1.77)
LOGC	22,154 (21,540–22,588)	84.8 (82.4–86.4)	19,143 (16,883–19,143)	52.6 (46.4–59.1)	−1.75 * (−1.83 to −1.66)
Central Latin America					
EOGC	2590 (2515–2663)	3.2 (3.1–3.3)	4707 (3923–5598)	3.6 (3.0–4.3)	0.25 * (0.16 to 0.35)
IOGC	1128 (1080–1176)	23.2 (22.2–24.2)	2239 (1859–2706)	17.3 (14.3–20.9)	−1.14 * (−1.22 to −1.05)
LOGC	11,778 (11,245–12,127)	86.5 (82.6–89.0)	23,563 (20,089–27,595)	59.0 (50.3–69.1)	−1.66 * (−1.77 to −1.55)
Central sub-Saharan Africa					
EOGC	544 (412–698)	2.2 (1.7–2.9)	905 (673–1187)	1.5 (1.1–1.9)	−1.54 * (−1.61 to −1.46)
IOGC	306 (224–406)	20.7 (15.1–27.4)	498 (353–678)	13.2 (9.4–18.0)	−1.59 * (−1.67 to −1.52)
LOGC	1875 (1552–2241)	49.4 (40.9–59.0)	2846 (2308–3538)	33.3 (27.0–41.3)	−1.46 * (−1.55 to −1.37)
East Asia					
EOGC	49,305 (42,924–56,328)	7.1 (6.2–8.2)	68,382 (56,611–81,595)	9.2 (7.6–10.9)	0.98 * (0.69 to 1.27)
IOGC	29,578 (25,111–34,639)	59.3 (50.3–69.5)	50,893 (41,026–62,257)	39.5 (31.8–48.3)	−1.30 * (−1.75 to −0.84)
LOGC	246,831 (217,625–277,992)	164.6 (145.1–185.4)	507,214 (428,407–597,285)	139.3 (117.6–164.0)	−0.32 (−0.67 to 0.03)
Eastern Europe					
EOGC	9948 (9280–10,298)	9.0 (8.4–9.3)	6053 (5379–6793)	6.2 (5.5–6.9)	−2.26 * (−2.66 to −1.87)
IOGC	9558 (9066–9932)	60.3 (57.2–62.6)	3525 (3098–4038)	27.2 (23.9–31.2)	−3.34 * (−3.66 to −3.02)
LOGC	67,527 (65,586–68,832)	138.1 (134.2–140.8)	44,541 (40,280–49,271)	72.0 (65.1–79.6)	−2.62 * (−2.81 to −2.42)
Eastern sub-Saharan Africa					
EOGC	1917 (1532–2213)	2.3 (1.8–2.7)	2711 (2257–3265)	1.4 (1.1–1.6)	−2.14 * (−2.27 to −2.01)
IOGC	868 (736–1004)	18.7 (15.8–21.6)	1127 (936–1341)	10.7 (8.9–12.7)	−2.16 * (−2.25 to −2.06)
LOGC	5449 (4774–6059)	44.8 (39.3–49.8)	7917 (7025–8913)	30.7 (27.2–34.6)	−1.44 * (−1.52 to −1.37)
High-income Asia Pacific					
EOGC	16,695 (16,188–17,190)	18.0 (17.4–18.5)	6408 (5628–7283)	7.9 (6.9–9.0)	−3.08 * (−3.23 to −2.92)
IOGC	9563 (9133–9981)	92.0 (87.9–96.0)	4235 (3569–4926)	31.2 (26.3–36.3)	−3.80 * (−3.97 to −3.64)
LOGC	97,474 (93,352–99,971)	278.6 (266.8–285.7)	117,525 (98,815–135,906)	169.8 (142.8–196.3)	−1.84 * (−1.90 to −1.77)
High-income North America					
EOGC	2493 (2423–2554)	1.7 (1.6–1.7)	2785 (2405–3259)	1.7 (1.4–2.0)	−0.17 (−0.39 to 0.05)
IOGC	1298 (1254–1343)	10.2 (9.8–10.5)	1872 (1578–2211)	8.0 (6.8–9.5)	−0.76 * (−0.84 to −0.68)
LOGC	26,528 (25,281–27,293)	45.6 (43.5–47.0)	32,927 (28,746–37,397)	30.4 (26.6–34.6)	−1.82 * (−1.96 to −1.67)
North Africa and Middle East					
EOGC	4441 (3843–4988)	2.7 (2.4–3.1)	7272 (6171–8484)	2.2 (1.8–2.5)	−0.84 * (−1.14 to −0.54)
IOGC	2226 (1779–2407)	20.7 (17.4–23.5)	3672 (3153–4263)	13.2 (11.3–15.3)	−1.53 * (−1.77 to −1.30)
LOGC	16,928 (14,683–18,502)	59.1 (51.2–64.5)	31,315 (28,294–34,373)	43.9 (39.7–48.2)	−0.90 * (−1.06 to −0.74)
Oceania					
EOGC	126 (95–158)	4.0 (3.0–5.0)	294 (212–392)	4.3 (3.1–5.8)	0.38 * (0.29 to 0.48)
IOGC	41 (29–56)	21.4 (15.4–29.1)	100 (69–139)	19.7 (13.5–27.4)	−0.19 * (−0.25 to −0.13)
LOGC	252 (200–309)	52.1 (41.3–63.8)	543 (423–679)	47.9 (37.3–59.8)	−0.40 * (−0.47 to −0.32)
South Asia					
EOGC	14,580 (12,912–16,083)	2.8 (2.4–3.0)	21,127 (18,261–24,233)	2.2 (1.9–2.5)	−0.81 * (−0.90 to −0.73)
IOGC	6075 (5300–6752)	16.8 (14.7–18.7)	9650 (8097–11,374)	12.1 (10.2–14.3)	−1.16 * (−1.44 to −0.88)
LOGC	36,919 (33,052–40,694)	39.6 (35.4–43.6)	68,622 (60,511–78,365)	29.2 (25.8–33.4)	−1.15 * (−1.23 to −1.06)
Southeast Asia					
EOGC	5693 (4810–6437)	2.4 (2.0–2.7)	6354 (5397–7407)	1.8 (1.5–2.0)	−1.32 * (−1.48 to −1.16)
IOGC	2728 (2339–3126)	17.2 (14.8–19.7)	3728 (3151–4364)	9.7 (8.2–11.4)	−2.05 * (−2.11 to −2.00)
LOGC	19,648 (16,960–21,782)	46.4 (40.0–51.4)	29,973 (26,787–33,361)	28.6 (25.6–31.9)	−1.92 * (−2.02 to −1.83)
Southern Latin America					
EOGC	845 (810–885)	3.5 (3.3–3.6)	916 (704–1197)	2.7 (2.1–3.5)	−0.88 * (−0.98 to −0.78)
IOGC	536 (500–576)	24.4 (22.7–26.2)	583 (425–790)	15.9 (11.6–21.5)	−1.31 * (−1.40 to −1.22)
LOGC	7075 (6817–7297)	89.3 (86.1–92.2)	9195 (7372–11,407)	65.0 (52.1–80.7)	−1.18 * (−1.27 to −1.09)
Southern sub-Saharan Africa					
EOGC	562 (514–608)	2.2 (2.0–2.3)	619 (522–728)	1.5 (1.2–1.7)	−1.75 * (−2.06 to −1.43)
IOGC	221 (196–250)	14.4 (12.8–16.4)	311 (262–372)	9.7 (8.1–11.5)	−1.32 * (−1.74 to −0.90)
LOGC	1636 (1465–1795)	36.3 (32.5–39.9)	2680 (2476–2912)	28.4 (26.3–30.9)	−1.03 * (−1.43 to −0.63)
Tropical Latin America					
EOGC	2493 (2410–2578)	3.2 (3.1–3.3)	3064 (2911–3226)	2.6 (2.4–2.7)	−0.73 * (−0.87 to −0.60)
IOGC	1417 (1351–1493)	26.8 (25.6–28.3)	1864 (1749–1990)	14.4 (13.5–15.3)	−2.06 * (−2.14 to −1.99)
LOGC	12,199 (11,659–12,610)	80.7 (77.1–83.4)	19,609 (18,270–20,572)	47.0 (43.8–49.3)	−1.99 * (−2.10 to −1.88)
Western Europe					
EOGC	6143 (6008–6270)	3.2 (3.1–3.2)	4612 (3924–5399)	2.4 (2.1–2.8)	−0.97 * (−1.13 to −0.81)
IOGC	4129 (3966–4295)	18.0 (17.3–18.7)	3601 (3031–4261)	11.2 (9.4–13.2)	−1.59 * (−1.66 to −1.53)
LOGC	83,680 (79,788–85,815)	86.1 (82.1–88.3)	78,242 (67,832–88,026)	54.1 (46.9–60.9)	−1.72 * (−1.79 to −1.65)
Western sub-Saharan Africa					
EOGC	1321 (1109–1516)	1.6 (1.3–1.8)	2389 (1933–2896)	1.1 (0.9–1.3)	−1.06 * (−1.13 to −0.99)
IOGC	742 (618–872)	14.2 (11.8–16.7)	1266 (1022–1552)	9.9 (8.0–12.2)	−1.07 * (−1.12 to −1.03)
LOGC	6465 (5634–7235)	45.0 (39.2–50.4)	11,307 (9863–12,868)	37.4 (32.6–42.5)	−0.43 * (−0.52 to −0.33)

CI, confidence interval; EAPC, estimated annual percentage change; EOGC, early-onset gastric cancer (15–49 years); IR, incidence rate; IOGC, intermediate-onset gastric cancer (50–54 years); LOGC, later-onset gastric cancer (≥55 years); UI, uncertainty interval. * Indicates statistical significance.

## Data Availability

The data generated and analyzed in this study are available from the Global Health Data Exchange query tool (http://ghdx.healthdata.org/gbd-results-tool) (accessed on 18 January 2022). The data that support the main findings of this study are also available from the corresponding author upon reasonable request.

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
