# Peer review of "Global, Regional, and National Burdens with Temporal Trends of Early-, Intermediate-, and Later-Onset Gastric Cancer from 1990 to 2019 and Predictions up to 2035"

_cancers, 2022, doi:10.3390/cancers14215417_

Round 1

Reviewer 1 Report

I have reade with interest the article entitled "Global, Regional, and National Burdens with Temporal Trends 2 of Early-, Intermediate-, and Later-onset Gastric Cancer from 3 1990 to 2019 and Predictions up to 2035", which analyses the incidence and death rates of gastric cancer at different ages of onset (early <50 yr-old, intermediate 50-54 yr-old, and late >55 yr-old) from 1990 to 2019 and from 2020 to 2035. The data presented are obtaied from 203 countries with different incidences of gastric cancer and sociodemographic index, accountinf for other clinical and pathological factors.

This manuscript is designed as an epidemiological study. The work is very complete and gives an general overview of the evolution and trends of this deadly type of carcinoma worldwide.

The figures and tables are fit to the information given and the supplementary material is exhaustive. 

Author Response

Reviewer #1

I have read with interest the article entitled "Global, Regional, and National Burdens with Temporal Trends 2 of Early-, Intermediate-, and Later-onset Gastric Cancer from 3 1990 to 2019 and Predictions up to 2035", which analyses the incidence and death rates of gastric cancer at different ages of onset (early <50 yr-old, intermediate 50-54 yr-old, and late >55 yr-old) from 1990 to 2019 and from 2020 to 2035. The data presented are obtained from 203 countries with different incidences of gastric cancer and sociodemographic index, accounting for other clinical and pathological factors.

This manuscript is designed as an epidemiological study. The work is very complete and gives a general overview of the evolution and trends of this deadly type of carcinoma worldwide.

The figures and tables are fit to the information given and the supplementary material is exhaustive. 

Response: Thank you for your positive and encouraging comments, and we appreciate your interests in our study.

Reviewer 2 Report

Dear authors and editor,

The manuscript titled ‘’Global, Regional, and National Burdens with Temporal Trends of Early-, Intermediate-, and Later-onset Gastric Cancer from 1990 to 2019 and Predictions up to 2035 ‘’ analyse the incidences, deaths, and trends of EOGC, IOGC, and LOGC(early, intermediate, and late-onset gastric cancer) from 1990 to 2019 at global, regional and national levels. The authors also predicted the future incidences, deaths and trends from 2020 to 2035.

It is worth emphasizing, that this study provides the first analyses of the burden, trends, and predicted trends in EOGC, IOGC, and LOGC worldwide. It is known from the literature that gastric cancer among younger patients represents a distinct disease from that in elderly. Gastric cancer among younger adults appears to be genetically and clinically distinct from conventional gastric cancer. We are observing more often diffuse histology, poor differentiation, more peritoneal metastasis and it is more common in young women. For the above-mentioned reasons, the subject of the project itself is very interesting and clinically necessary

The study revealed differences in the burdens and trends of EOGC, IOGC and LOGC, especially in patients with EOGC. The authors suggest that the incidence rates of EOGC and LOGC will increase worldwide from 2020 to 2035. Whereas the estimations from the 2019 were consistent with previous studies.

The paper is well-organised, the language is correct and the content is understandable. Statistical tests mostly correctly selected and performed. Literature properly selected and up to date(almost 100% citations from the last 10 years). I believe they add some contribution to the literature.

However the manuscript is good, I have some comments that should be clarified.

1. There are different classifications of gastric cancers according to the age of onset of the cancer. Most often, early onset gastric cancer is at the age of 45 or younger. For what reason did the authors recognize 49 years as the border?

2. Clinicopathological, Somatic, and Survival Analyses among EOGC, IOGC, and LOGC

Many factors influence the survival of patients after GC surgery. As is known, the histological type, the stage of GC including the number of lymph nodes involved, the type of surgery and the type of chemotherapy. In addition, in the elderly patients with GC, the course of the disease is usually slow. For the above-mentioned reasons, in my opinion this paragraph is too poor, therefore it should be expanded or completely remove.

I support the publication of the manuscrypt.

Thank you for your choice me as a reviewer.

Author Response

Reviewer #2

The manuscript titled ‘’Global, Regional, and National Burdens with Temporal Trends of Early-, Intermediate-, and Later-onset Gastric Cancer from 1990 to 2019 and Predictions up to 2035‘’ analyze the incidences, deaths, and trends of EOGC, IOGC, and LOGC(early, intermediate, and late-onset gastric cancer) from 1990 to 2019 at global, regional and national levels. The authors also predicted the future incidences, deaths and trends from 2020 to 2035.

It is worth emphasizing, that this study provides the first analyses of the burden, trends, and predicted trends in EOGC, IOGC, and LOGC worldwide. It is known from the literature that gastric cancer among younger patients represents a distinct disease from that in elderly. Gastric cancer among younger adults appears to be genetically and clinically distinct from conventional gastric cancer. We are observing more often diffuse histology, poor differentiation, more peritoneal metastasis and it is more common in young women. For the above-mentioned reasons, the subject of the project itself is very interesting and clinically necessary

The study revealed differences in the burdens and trends of EOGC, IOGC and LOGC, especially in patients with EOGC. The authors suggest that the incidence rates of EOGC and LOGC will increase worldwide from 2020 to 2035. Whereas the estimations from the 2019 were consistent with previous studies.

The paper is well-organized, the language is correct and the content is understandable. Statistical tests mostly correctly selected and performed. Literature properly selected and up to date (almost 100% citations from the last 10 years). I believe they add some contribution to the literature. However, although the manuscript is good, I have some comments that should be clarified.

Response: Thank you for your positive and encouraging comments, and we appreciate your interests in our study.

  1. There are different classifications of gastric cancers according to the age of onset of the cancer. Most often, early onset gastric cancer is at the age of 45 or younger. For what reason did the authors recognize 49 years as the border?

Response: Thank you for your important comments.  We agree with you that the threshold for classifying early-onset gastric cancer varies among different study. We listed some recent studies that applied the age of 49 years as the border for your references, as follows:

1.Tavakkoli A, Pruitt SL, Hoang AQ, Zhu H, Hughes AE, McKey TA, Elmunzer BJ, Kwon RS, Murphy CC, Singal AG. Ethnic Disparities in Early-Onset Gastric Cancer: A Population-Based Study in Texas and California. Cancer Epidemiol Biomarkers Prev. 2022;31(9):1710-1719.

2.Ugai T, Sasamoto N, Lee HY, Ando M, Song M, Tamimi RM, Kawachi I, Campbell PT, Giovannucci EL, Weiderpass E, Rebbeck TR, Ogino S. Is early-onset cancer an emerging global epidemic? Current evidence and future implications. Nat Rev Clin Oncol. 2022;19(10):656-673. 

3.Pocurull A, Herrera-Pariente C, Carballal S, Llach J, Sánchez A, Carot L, Botargues JM, Cuatrecasas M, Ocaña T, Balaguer F, Bujanda L, Moreira L. Clinical, Molecular and Genetic Characteristics of Early Onset Gastric Cancer: Analysis of a Large Multicenter Study. Cancers (Basel). 2021;13(13):3132.

The above studies, authors applied 49-year as the threshold for classifying early-onset gastric cancer.  We hope this may address your concerns. Thank you for this important comment.

  1. Clinicopathological, Somatic, and Survival Analyses among EOGC, IOGC, and LOGC

Many factors influence the survival of patients after GC surgery. As is known, the histological type, the stage of GC including the number of lymph nodes involved, the type of surgery and the type of chemotherapy. In addition, in the elderly patients with GC, the course of the disease is usually slow. For the above-mentioned reasons, in my opinion this paragraph is too poor, therefore it should be expanded or completely remove.

Response: Thank you for your important comments, and we fully agree with you that this part should be completely removed.  We have removed this part as you suggested.

I support the publication of the manuscrypt. Thank you for your choice me as a reviewer.

Response: Thank you for your positive comments and interests in our study.

Round 2

Reviewer 1 Report

The manuscript may be published in its actual form.